# Design and Mechanical Properties of Maximum Bulk Modulus Microstructures Based on a Smooth Topology with Grid Point Density

**Xin Zhou [1], Chenglin Tao [1,*], Xi Liang [1,2,*], Zeliang Liu [1,2]** and **Huijian Li [1,2]**

[1] School of Civil Engineering and Mechanics, Yanshan University, Qinhuangdao 066004, China; lxzhouxin@126.com (X.Z.)
[2] Hebei Key Laboratory of Mechanical Reliability for Heavy Equipments and Large Structures, Yanshan University, Qinhuangdao 066004, China
* Correspondence: taochenglin@stumail.ysu.edu.cn (C.T.); ysulxi@163.com (X.L.)

**Abstract:** The aim of topology optimisation is to determine the optimal distribution of material phases within the periodic cells of a microstructure. In this paper, the density of grid points under element volume fraction is constructed to replace the finite elements in the traditional SIMP framework, avoiding jagged and blurry boundaries in the computational process due to grid dependence. This is then combined with homogenisation theory, a microstructure topology optimisation algorithm with maximum bulk modulus under prescribed volume constraints is proposed, which can obtain 2D and 3D topologies with smooth boundaries. In addition, a closed form expression for the two-dimensional topological concave edge structure (taking the most typical topology as an example) was derived, and a compression experiment was conducted on the topological microstructure based on 3D metal printing technology. Scanning electron microscopy showed that the powder bonded on the surface of the printed structure was not completely melted and the step effect caused the finite element analysis results to be higher than the experimental results. Overall, the finite element simulation and experimental results of the concave surface structure have good consistency, with high strength and energy absorption effects. Topologies based on grid point density obtain microstructures with smooth boundaries, and the introduction of the Heaviside smoothing function and multiple filtering steps within this algorithm leads to more robust optimisation, facilitating 3D or 4D printing of microstructures that meet specific design requirements and confirming the feasibility of the proposed topology for lightweighting studies.

**Keywords:** microstructures; homogenisation; bulk modulus; topology optimisation; 3D printing





## 1. Introduction

Large-scale computing and advanced 3D printing manufacturing techniques enable the design of microstructures with fine and complex geometrical features [1]. Examples include cell and lattice structures, which are increasingly being investigated for their high stiffness-to-weight and strength-to-weight ratios, as well as their excellent energy absorption and sound insulation properties [2–5], and are used in a range of lightweight structures, including in the aerospace, automotive, and military industries. Topology optimisation provides an efficient computational method to find the optimal material distribution in the design domain under specific constraints and to obtain a wide range of geometrical designs for cellular structures. Various topology optimisation algorithms have been proposed based on different strategies, such as isotropic microstructure topology optimisation with penalty (SIMP) [6–8], evolutionary structure algorithms (ESO) [9–11], and the level set method (LSM) [12,13].

The SIMP method is the most common penalty method in pseudo-density-based topology optimisation algorithms; first, the design domain is discretised into a series of

cells; when the density is equal to 1, it represents solid material and when the density is equal to 0, it represents hole material. The high efficiency and compactness of the algorithm has led to the SIMP method being widely studied and applied to solve related problems in various fields, such as dynamics [14,15], electromagnetic fields [16,17], acoustics [18–20], and piezoelectric structures [21,22]. However, this method suffers from the presence of jagged edges in the topologically optimised structures obtained, and the small number of density values between 0 and 1, which inevitably produce blurry boundaries that require a post-processing mechanism to be introduced before it can be applied [23,24]. Increasing the number of elements is an easy way to obtain smooth boundaries, but this increases the computational cost, especially in the topology optimisation of 3D structures. In addition, pseudo-density fields based on non-uniform rational basis spline curves (NURBS) to describe the continuum topology [25,26] were used to unify the construction of CAD geometry models and CAE analysis models, enabling seamless integration of CAD and CAE information, and obtaining clear boundaries. Gao et al. [27] proposed a NURBS-based finite cell method for topology optimisation of the design domain of any clipped shape in a structure, and the tessellation problem generated in the SIMP framework was avoided by applying a filtering method. Qian et al. [28] used the method of embedding shapes in the B spline parameter space to avoid the tessellation phenomenon by using the density corresponding to the control vertices of the spline curve as a design variable. Da et al. [29] provide an evolutionary topology optimisation (ETO) algorithm based on volume fractions, where a level set function established by node sensitivity performs the smoothing design of the boundary. Fu et al. [30,31] used the SIMP method framework to perform topological design of macroscopic structures using element volume fractions, achieving smooth boundaries of grid points. Ullah et al. [32,33] integrated Bidirectional evolutionary structural optimisation (BESO), the level set method, and non-uniform rational B splines (NURBS) to obtain topologies with smooth boundaries. These above works mainly focus on the topology optimisation design of 2D and 3D macroscopic structures with maximum stiffness as the goal. This paper, meanwhile, combines the expression of boundary smoothing in ETO and homogenisation theory and carries out density based material (hollow/solid) optimisation design for element grid points to achieve topology optimisation of cellular structures in order to find an optimised configuration that satisfies specific design requirements, such as maximum stiffness, maximum shear stiffness [34], negative Poisson's ratio [35], etc.

In terms of microstructural design, the lattice structure is often considered to be homogeneous, and it can achieve excellent mechanical and other properties through the combination of several periodic microstructures following specific patterns, while topological microstructures that can be easily assembled can be achieved using homogenisation theory [36]. Huang et al. [37] proposed a periodic microstructure design with maximum volume and shear modulus based on BESO and homogenisation theory. A variety of conventional periodic lattice structures have emerged, such as chiral structures [38,39], inner-concave structures [40,41], star structures [42,43], rotating polygons [44,45], etc. In the analysis of the mechanical properties of such lattice structures, Alderson et al. [46] used finite element and experimental methods to obtain in-plane elastic constants for chiral and anti-chiral lattices. Lira et al. [47] described the transverse shear modulus of multi-inclusion honeycomb structures and Harkati et al. [48] solved for the in-plane elasticity of a new type of multi-concave expanded honeycomb structure. In addition, by changing the configuration and parameters of the microstructure, the conventional structure can achieve variable properties, such as enhanced stiffness, while maintaining the original mechanical properties [49]. The novel structures obtained by topology optimisation methods provide new inspiration for the design of lattices. Using the topological structure as a basis for design, the lattice structure can also be stiffness adjustable after adjusting the design parameters, providing a basis for the secondary design of topology optimised novel structures. Therefore, finding an efficient method for solving the expressions for the elastic constants

of microstructures after parameter normalisation is an important step in the evaluation and design of new structures.

In terms of microstructural applications, the filling design of microstructures can optimise the mechanical properties of macrostructures. For example, Liu et al. [50] designed 2D microstructures with controlled pore volume and distribution, which were filled based on the mechanical properties of a single cytosolic element. Zhao et al. [51,52] artificially introduced a new type of cubic cytosolic element and designed a microstructural distribution framework, which significantly improved the elastic modulus of the structure compared to the filling of a single structure. The topology optimisation algorithm provides a wider range of design ideas and is a powerful and free design tool. The optimised design from the Matlab platform is input into the finite element platform and transformed into solid structural features, which are periodically arranged structures formed by arrays. This paper takes this as a discussion object to analyse the mechanical properties of the topology-optimised microstructures after their combination. The research on topology optimisation of microstructures mostly focuses on the design and numerical simulation verification stages. Currently, additive manufacturing technology supplements the difficult-to-process topology optimisation of microstructures. Among them, Robbins et al. [53] demonstrated the correlation between topology optimisation and homogenisation principles in designing cell structures, ultimately achieving printed products based on adhering to minimum microstructure topology optimisation. Belhabib et al. [54] used melt deposition modeling techniques to print the microstructure of the topology and conducted compression tests until densification to evaluate the design.

The topological optimisation of topological 2D cytomel structure under homogenisation theory is carried out by introducing the method of grid point density, solid or hollow design for the grid points assigned to each element, constructing a smooth boundary, and solving the elastic constants of the structure by topologically obtaining a typical 2D internally concave curved-edge structure as an example. Topology optimisation of 3D cellular structure is carried out. Based on 3D printing technology, compression experiments and finite element analysis are carried out on the topological 3D concave surface structure, and the compression performance of the structure is discussed.

## 2. Homogenised Microstructures Topology with Smooth Boundaries

### 2.1. Setting of Grid Point Density

To solve the problems of jagged shapes and boundary blurring that occur during topology optimisation of cells, the finite element mesh is uniformly discretised into n grid points, and the grid points of each element are designed to be solid/hollow to form clear topological boundaries. To involve the grid point density in the FEA calculations, the volume fraction is expressed in terms of the grid point density $\rho_{e,i}$, which is:

$$\Phi_e = \frac{1}{N}\sum_{i=1}^{N}\rho_{e,i},\ \rho_{e,i} \in \{0,\ 1\} \tag{1}$$

where, $\rho_{e,i}$ is the eth grid point within the *i*th element, $\rho_{e,i} = 1$ indicates that the grid point is solid, $\rho_{e,i} = 0$ indicates that the grid point is hollow. $\Phi_e$ is the volume fraction of the eth element, $N$ is the total number of grid points in each element, $\Phi_e = 1$ indicates that all grid points within the cell are solid elements, $\Phi_e = 0$ indicates that all grid points within the cell are hollow elements, $0 < \Phi_e < 1$ indicates that the cell has both solid and hollow elements, is the model boundary location of the cells, as shown in Figure 1. Based on the SIMP framework, the Young's modulus of a cell is defined as:

$$E(\Phi_e) = (1 - \Phi_e^p)E_{\min} + \Phi_e^p E_0 \tag{2}$$

where, $E_0$ is the Young's modulus of the solid material, and $E_{\min}$ is the porous material.

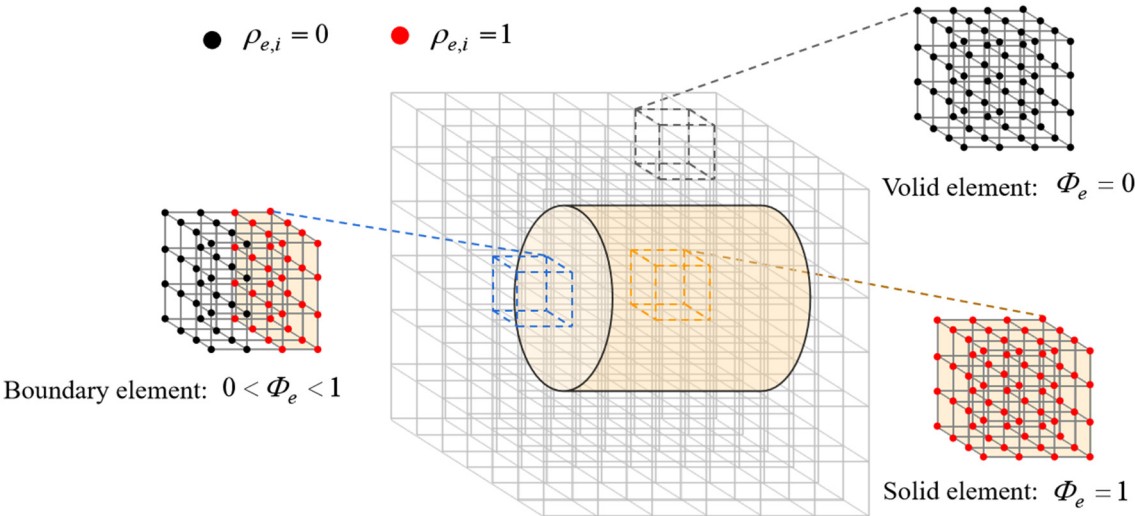

**Figure 1.** Density of grid points within elements under smooth boundaries.

### 2.2. Homogenisation Theory

Homogenisation theory predicts the equivalent properties of the macrostructure from the properties of the microstructure, as shown in Figure 2, within the global coordinate system $X_1 - X_2$ and the microstructure located in a periodic repetitive arrangement under the local coordinate system $y_1 - y_2$.

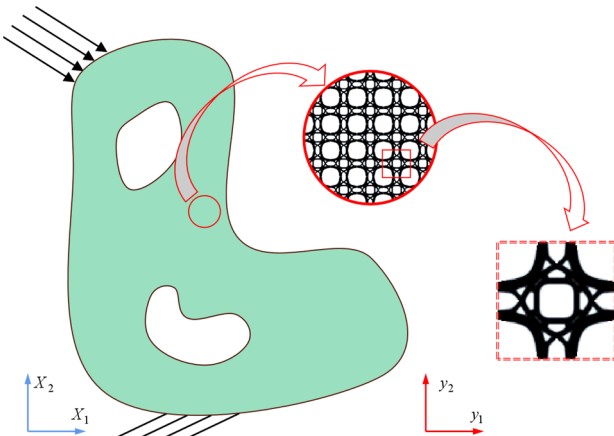

**Figure 2.** 2D macrostructure and microstructure under periodic boundary conditions.

Progressive expansion of macro-displacement field $\chi^\varepsilon$ is:

$$\chi^\varepsilon(X) = \chi_0(X, y) + \varepsilon\chi_1(X, y) + \varepsilon^2\chi_2(X, y) \cdots \varepsilon^n\chi_n(X, y), \ y = X/\omega \tag{3}$$

The function depends on global macro variables and local macro variables, where $\varepsilon$ is the method factor for scaling macroscopic dimensions by microstructural dimensions. When $\varepsilon << 1$, the dependence on $y$ can be considered periodic for a fixed macroscopic point $X$. When considering only the gradually expanding first order terms in the above equation, the homogenised stiffness tensor:

$$E_{ijkl}^H = \frac{1}{|\Omega|} \int_\Omega E_{ijpq}(\varepsilon_{pq}^{0(kl)} - \varepsilon_{pq}^{*(kl)})d\Omega \tag{4}$$

where, $\Omega$ denotes the area of the microstructure (representing the volume of structure in 3D topology). $i, j, k, l = 1, 2, \cdots, d$ denotes the index vector value, $\varepsilon_{pq}^{0(kl)}$ denotes the

predefined macroscopic strain field. Contains three separate vectors in two dimensions and six separate vectors in three dimensions. $\varepsilon_{pq}^{*(kl)}$ is the locally varying strain field caused by the application of $\varepsilon_{pq}^{0(kl)}$ to the structure, the sensitivity equation for the homogenised elastic tensor is obtained by applying the unit test strain directly to the boundary of the unit, as shown below:

$$\frac{\partial E_{ijkl}^H}{\partial \Phi_e} = \frac{1}{|V|} p \sum_{i=1}^{N} \rho_{e,i}^{p-1} (E_0 - E_{\min}) \left(u_e^{(ij)}\right)^T k_1 u_e^{(kl)} \tag{5}$$

where, $V$ is the volume of the 2D microstructure, $u_e^{(ij)}$ is the perceived displacement field within the cell, corresponding to the element test strain field, and $k_e$ is element stiffness matrix. The model is solved using the Moving $d$ Asymptotic Method (MMA) criterion.

*2.3. Optimisation Model*

Taking $c(E_{ijkl}(\Phi_e))$ as the general form of the objective function, the mathematical formulation of the optimisation problem is expressed as:

$$\min : \ c(E_{ijkl}(\Phi_e)) \tag{6}$$

$$\text{s.t} : \begin{cases} KU^{A(kl)} = F^{(kl)}, & k,l = 1,\dots,d \\ \sum_{e=1}^{n} v_e \Phi_e / |\Omega| \leq \vartheta \\ 0 \leq \Phi_e \leq 1, & e = 1,\dots,M \end{cases} \tag{7}$$

where, $K$ is the overall stiffness matrix, $U^{A(kl)}$ and $F^{kl}$ are the overall displacement vectors and external force components, respectively, $\Omega$ is the spatial dimension, $v_e$ is the volume of the element, $\vartheta$ is the upper limit of the volume fraction, and $M$ is the number of finite element elements. the objective function is a function of the uniform stiffness tensor.

*2.4. Evolution of Smooth Boundaries*

A quadratic filtering method [30] is introduced on top of SIMP, where the filtered cell volume fraction is transferred to the nodes by means of a filter, and then the resulting node densities are assigned to the grid points using a shape function, and the grid point densities within the cells are obtained by means of linear interpolation of the node densities based on the tanh's Heaviside smoothing function [31], in order to form a smooth boundary.

**3. 2D Topology Optimisation with Maximised Bulk Modulus**

*3.1. Numerical Results*

The topologic ptimizationion of a 2D model is designed with the maximum bulk modulus objective property, where the elastic modulus of the solid and the hole are 1 and $1 \times 10^{-9}$, respectively, and the Poisson's ratio is 0.3. The 2D model is discretised using a $100 \times 100$ finite element mesh. The convergence condition is defined as the termination when the difference between adjacent objective functions is less than 0.001 in absolute value or when the maximum iteration step is 150 steps. The discrete 2D environment is defined as 11→1, 22→2, 12→3. The representation of elastic tensor is obtained as:

$$E^H = \begin{bmatrix} E_{1111}^H & E_{1122}^H & E_{1112}^H \\ E_{1122}^H & E_{2222}^H & E_{2212}^H \\ E_{1211}^H & E_{1112}^H & E_{1212}^H \end{bmatrix}_{2D} \tag{8}$$

Expression of the objective function of the maximum bulk modulus: $c(E_{ijkl}(\rho)) = 4K = E_{11}^H + E_{22}^H + 2E_{12}^H$.

As shown in Figure 3, a series of 2D topological microstructures under the maximisation of bulk modulus obtained by the change of initial structure [55] using the SIMP method, and a series of 2D topological microstructures under the maximisation of bulk

modulus obtained, can only be artificially deleted by deleting points on the fuzzy boundary. The use of fuzzy structural boundaries is the production of the topological structure and to obtain a better application. However, it cannot satisfy the topological requirements of boundary conditions on the volume fraction during process. Moreover, the jaggedness of the boundary leads to the appearance of step forms at the edges in the printing. The stress concentration phenomenon can easily occur at these positions, which effects the 3D printing process.

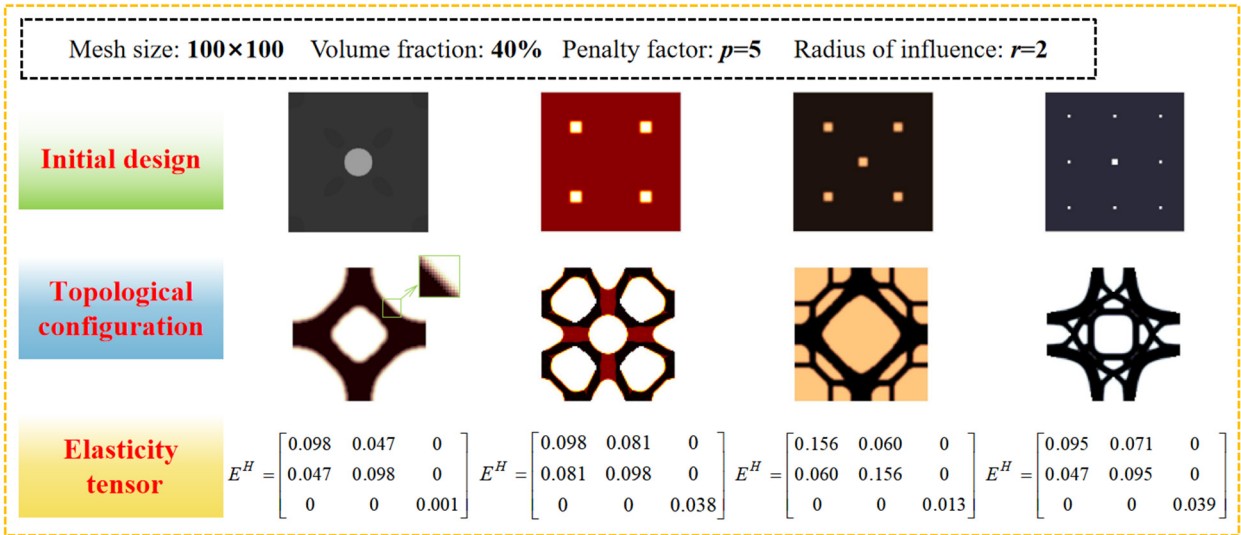

**Figure 3.** 2D topological microstructure with maximum bulk modulus (SIMP).

Based on the topological method proposed in this paper, four typical two-dimensional structures with clear boundaries are obtained through varying initial designs (the initial structure design is the same as that in Figure 3), as shown in Figure 4. It can be noted that under the same objective function, the structures of the topologies all appear in the form of more continuous concave curves, which are obtained by iterating the objective function at the maximum bulk modulus. Comparing the topology optimisation results shown in Figure 3, this method can effectively solve the checkerboard effect appearing in the SIMP method and make the structure boundary clear and smooth, which is more conducive to 3D printing without the need to modify the structure of the topology.

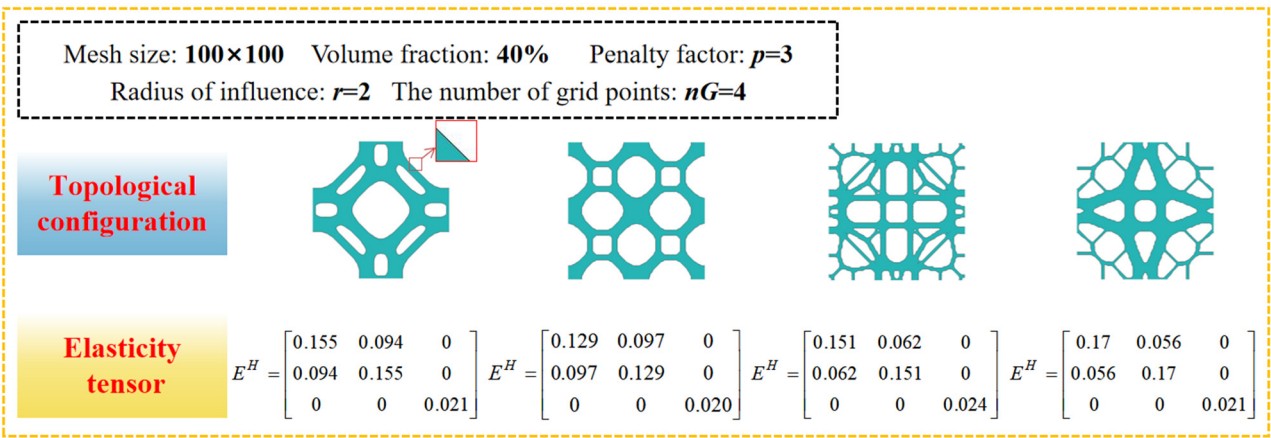

**Figure 4.** 2D topology after boundary clarification.

Three types of microstructures with special properties were obtained under the initial design with one hole, as shown in Figure 5. As shown in the figure, the two-dimensional

metamaterial topology based on grid density has obtained the optimal solution within 150 steps. This algorithm has high iterative stability and fast optimisation efficiency. The iterative process is relatively stable and can achieve the optimal topology of microstructures within a limited number of iteration steps, indicating the feasibility and authenticity of the algorithm for microstructure topology. The ability of level set functions and MMC based methods to form smooth and clear boundaries is their unique advantage. However, the solution obtained by the level set method depends significantly on the initial guess design and regularisation techniques, and its convergence highly depends on grid discretisation and regularisation [56]. The MMC based method obtains clear and smooth boundaries, but local non-smooth points will appear at the intersection of components [57]. Compared to other topology optimisation algorithms, this algorithm is easier to combine with SIMP based methods in order to achieve specific goals.

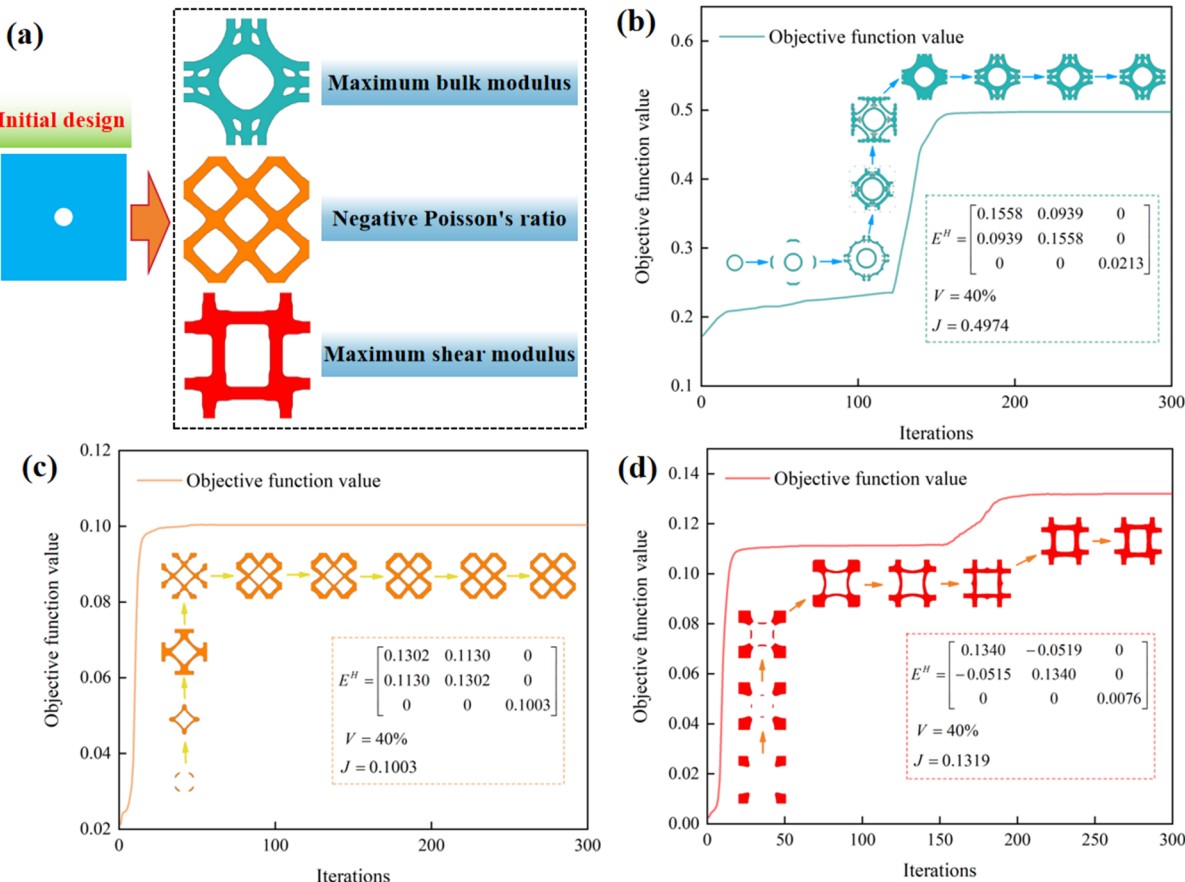

**Figure 5.** (**a**) 2D microstructure topology design. (**b**) Maximum stiffness microstructure topology convergence process. (**c**) Maximum shear modulus microstructure topology convergence process. (**d**) Negative Poisson's ratio microstructure topology convergence process.

The iterative process of two types of microstructures with similar configurations based on SIMP and grid point density topology is shown in Figure 6. The convergence of the SIMP algorithm is relatively slow, resulting in numerical instability in the optimisation process.

As shown in Table 1, the above types of microstructure topology optimisation algorithms can all achieve optimal solutions within a certain iteration step.

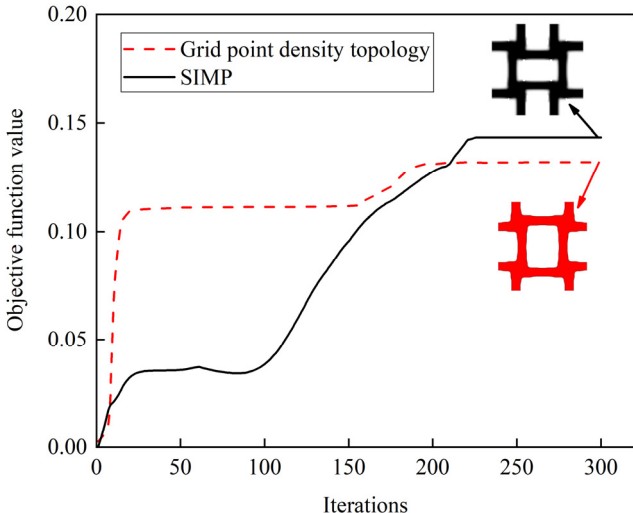

**Figure 6.** Iteration process.

**Table 1.** Iterations under different algorithms.

| Graphical | Algorithm | Iterations |
|:---:|:---:|:---:|
|  | SIMP with homogenisation | 150 [58] |
|  | SIMP with strain energy-based homogenisation method | 140 [59] |
|  | level set method | 98 [60] |
|  | Grid point density topology | 104 |

### 3.2. In-Plane Elastic Properties of 2D Topological In-Concave Curved-Edge Structures

#### 3.2.1. Equivalent Modulus of Elasticity

With the maximum bulk modulus as the target, the structures shown in Figures 3 and 4 are obtained. From the Figures, it can be seen that all these types of structures have similar internal concave properties. In this paper, the most typical 2D internal concave curved-edge structure is selected and the closed-form expression for the elastic constants in its face is solved, as shown in Figure 7. Figure 7a shows the maximum stiffness topological microstructure under uniaxial stress in the y-direction after a periodic arrangement. A simplification of the microstructure is shown in Figure 7b, which contains a topologically optimised structure with four curved edges and four connected semi-ligamentous beams, where $F$ can be expressed as a function of the applied stress. When $F = \sigma_y H$, considering the symmetry of the structure, only the upper side portion of the microstructure is analysed. While satisfying the above equilibrium conditions, the unknown forces and moments acting on the upper side of the microstructure can be obtained, giving $M = Fr\cos\theta/4$, where,

$r = \sqrt{h^2 + l^2}/2$, $\theta$ is the angle between the tangent line of the curved beam node and the axis. The strain energy of the microstructure is expressed as:

$$U = 2\frac{F^2(H/2 - h/2)}{2EA} + 4\frac{(F\sin\theta/2)^2 r}{2EA} + 2\int_0^r \frac{(Fr\cos\theta/4 - xF\cos\theta/2)^2}{2EI}dx \quad (9)$$

where, $E$ is the Young's modulus of the microstructure, $A$ is the cross-sectional area (for a rectangular cross-section of unit depth, $A = t$, $I = t^3/12$), giving a total displacement in the y-direction as:

$$\delta_y = \frac{\partial U}{\partial F} = \frac{F(H - h + r\sin^2\theta)}{EA} + \frac{Fr^3\cos^2\theta}{24EI} \quad (10)$$

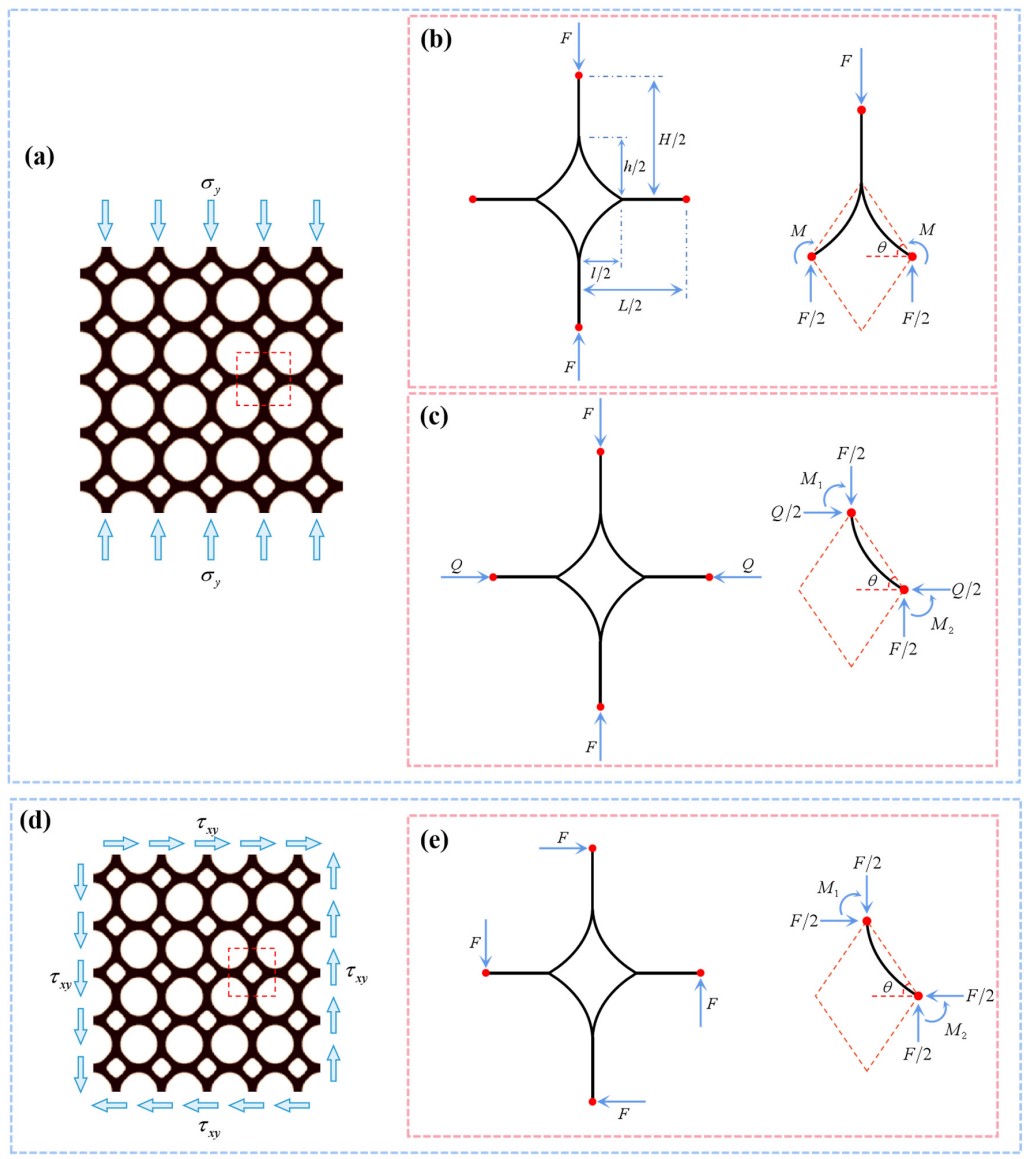

**Figure 7.** (**a**) Schematic diagram of a 2D topological inner concave curved-edge structure under uniaxial loading in the y-direction. (**b**) Schematic diagram of a 2D topological inner concave curved-edge structure under x-y shear loading. (**c**) Schematic diagram of a single cell element under uniaxial loading. (**d**) Schematic diagram of a single cell element subjected to virtual forces. (**e**) Schematic diagram of a single cell element subjected to shear loading.

In order to solve for the Poisson's ratio of the microstructure, when $F = 0$, the total displacement in the x-direction, the strain in the y-direction can be given as $\varepsilon_y = \delta_y / H$, and the equivalent modulus of elasticity is defined as:

$$E_y = \frac{tE/H}{1 - (h - r\sin^2\theta)/H + (r/h)^3 \cos^2\theta/2(t/H)^2} \tag{11}$$

### 3.2.2. Equivalent Poisson's Ratio

In order to calculate the Poisson's ratio of the cellular, a pair of virtual forces $Q$ acting on the cell in the x-direction is considered, as shown in Figure 7c, and a curved beam of a cellular structure is taken for analysis to obtain the unknown moments $M_1$ and $M_2$ at the ends of the curved edges. The expressions for $M_1$ and $M_2$ are $M_1 = Fr\cos\theta/4 - Pr\cos\theta/4$ and $M_2 = Pr\cos\theta/4 - Fr\cos\theta/4$, respectively. Thestrain energy of the cellular structure under the combined effect of virtual and external forces can be shown as:

$$U = 2\frac{F^2(H/2 - h/2)}{2EA} + 2\frac{P^2(L/2 - l/2)}{2EA} + 4\frac{[(F+P)\sin\theta/2]^2 r}{2EA} + 4\int_0^r \frac{[(P-F)(r\cos\theta/4 - x\cos\theta/2)]^2}{2EI}dx \tag{12}$$

The displacement in the direction of the virtual force is obtained from the second theorem of the cartesian to satisfy: $\delta_x = \partial U/\partial P|_{P=0}$. In this case, the displacement in the x direction can be shown as:

$$\delta_x = \frac{Fr\sin^2\theta}{EA} - \frac{F\cos\theta(3r^2 - 2r^3)}{12EI} \tag{13}$$

The stresses and strains in the microstructure in the x-direction are: $\sigma_x = F/L$, $\varepsilon_x = \delta_x/L$. Defining the equivalent Poisson's ratio of a structure as the negative of the ratio of the mean strain in the y-direction to the mean strain in the x-direction, and obtain the results:

$$\nu_{xy} = \frac{2t^2(r/L)\sin^2\theta - (r/L)(6r - 2r^2)\cos\theta}{2t^2(1 - h/H + (r/H)\sin^2\theta) + (r/H)r^2\cos^2\theta} \tag{14}$$

### 3.2.3. Equivalent Shear Modulus

To determine the shear modulus, as shown in Figure 7d, a uniform far-field shear stress $\tau_{xy}$ is applied to the lattice structure in this paper, and the microstructure is shown in Figure 7e. Four shear forces act on the external cut point of the cell, which can be determined as a function of the applied stress: $F = \tau_{xy}H$, $F' = \tau'_{xy}L$. Considering a curved beam of the microstructure, the components of the unknown forces and moments acting on the external cut point of this part can be determined as a function of $F$, $F'$: $M_1 = F/2(L/2 - l/2)$, $M_2 = F/2(H/2 - h/2)$. Thus, the strain energy of the microstructure is shown as:

$$U = 2\int_0^{H/2 - h/2} \frac{(Fx)^2}{2EI}dx + 2\int_0^{L/2 - h/2} \frac{(F'x)^2}{2EI}dx + 2\frac{(2F\sin\theta)^2 r}{2EA} + 2\frac{(2F'\sin\theta)^2 r}{2EA} + 4\int_0^{r_1} \frac{(F'/2(L/2 - l/2))^2}{2EI}dx + 4\int_0^{r_2} \frac{(F/2(H/2 - h/2))^2}{2EI}dx \tag{15}$$

where, $r_1 + r_2 = r$ , then $(\partial U/\partial F')/L$, $(\partial U/\partial F)/H$ gives the total change in shear strain $\gamma_{xy}$ between two lines initially parallel to the $x$ and $y$ axes. Finally, the equivalent shear modulus $G_{xy}$ of the structure is defined as the ratio of the average shear stress $\tau_{xy}$ to the average shear strain $\gamma_{xy}$, obtained as:

$$G_{xy} = \frac{t^3 E}{[(H-h)^3 + (L-l)^3] + 3[(H-h)^2 r_1 + (L-l)^2 r_2] + 16rt^2\sin^2\theta} \tag{16}$$

If $H = L$ , then

$$G_{xy} = \frac{t^3 E}{2[(H-h)^3] + 3[(H-h)^2\sqrt{2}h] + 16\sqrt{2}ht^2\sin^2\theta} \tag{17}$$

## 4. 3D Topology Optimisation and Experimental Analysis under Volume Modulus Maximisation

### 4.1. Numerical Results

The 3D model uses a $30 \times 30 \times 30$ finite element mesh, defined in a discrete 3D environment: $11\rightarrow1, 22\rightarrow2, 33\rightarrow3, 12\rightarrow4, 23\rightarrow5, 31\rightarrow6$. The representation of elastic tensor is obtained as:

$$E^H = \begin{bmatrix} E^H_{11} & E^H_{12} & E^H_{13} & E^H_{14} & E^H_{15} & E^H_{16} \\ E^H_{21} & E^H_{22} & E^H_{23} & E^H_{24} & E^H_{25} & E^H_{26} \\ E^H_{31} & E^H_{32} & E^H_{33} & E^H_{34} & E^H_{35} & E^H_{36} \\ E^H_{41} & E^H_{42} & E^H_{43} & E^H_{44} & E^H_{45} & E^H_{46} \\ E^H_{51} & E^H_{52} & E^H_{53} & E^H_{54} & E^H_{55} & E^H_{56} \\ E^H_{61} & E^H_{62} & E^H_{63} & E^H_{64} & E^H_{65} & E^H_{66} \end{bmatrix}_{3D} \tag{18}$$

Define the objective function for the 3D microstructure topological optimisation according to the maximum bulk modulus: $c(E_{ijkl}(\rho)) = 9K = \sum\limits_{i,j=1}^{3} E^H_{ij}$. Several types of bulk modulus maximising 3D microstructures were obtained, as shown in Figure 8a. The first type of 3D inner concave surface structure is used as an example for analysis in this paper, where the 3D maximum volume modulus topologically optimised structures with different volume fractions are shown in Table 2. For the traditional SIMP algorithm, the topology optimisation algorithm, based on grid point density, yields microstructures with smooth and clear boundaries, but the two types of optimised structures are consistent from the overall configuration, verifying the validity of the topologically optimised model with smooth boundaries. In addition to this, the algorithm can also obtain optimised designs for 3D topology based on different specific requirements, such as maximum shear modulus (Figure 8b), negative Poisson's ratio (Figure 8c), etc.

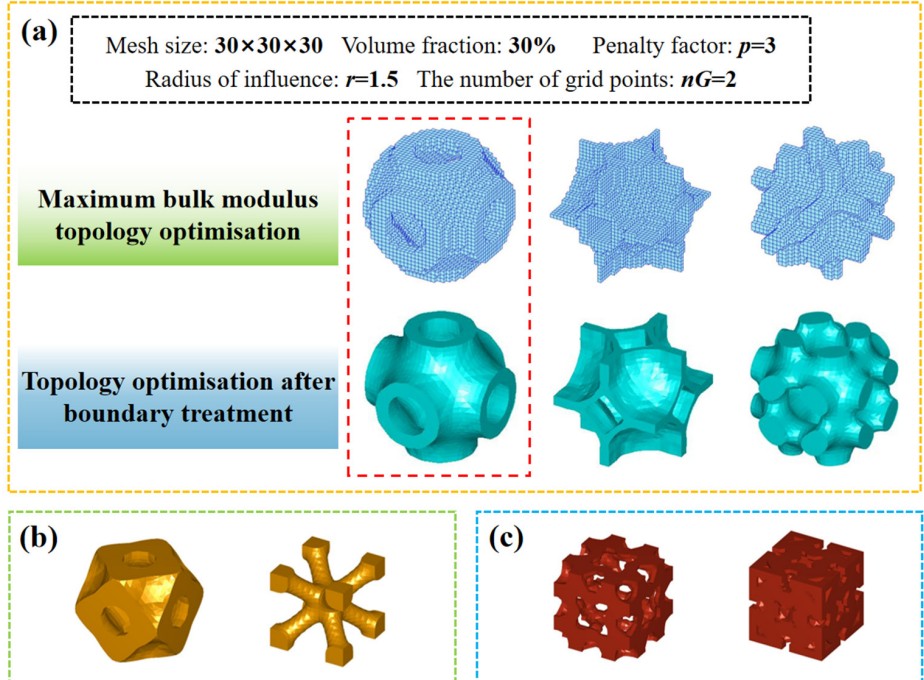

**Figure 8.** (**a**) 3D topology optimization of maximum bulk modulus. (**b**) 3D topology optimization of maximum shear modulus. (**c**) 3D topology optimization of Negative Poisson's ratio microstructure.

**Table 2.** Numerical optimisation results for a 3D internal concave surface structure.

| Volume Fraction | Topology Optimisation Model | 3D Microstructure | $E^H$ | 3D Surfaces of Elastics Modulus |
|---|---|---|---|---|
| 30% | SIMP |  | $\begin{bmatrix} 1.34 & 0.80 & 0.80 & 0 & 0 & 0 \\ 0.80 & 1.34 & 0.80 & 0 & 0 & 0 \\ 0.80 & 0.80 & 1.34 & 0 & 0 & 0 \\ 0 & 0 & 0 & 0.57 & 0 & 0 \\ 0 & 0 & 0 & 0 & 0.57 & 0 \\ 0 & 0 & 0 & 0 & 0 & 0.57 \end{bmatrix}$ |  |
| | Grid point density |  | $\begin{bmatrix} 3.17 & 1.80 & 1.80 & 0 & 0 & 0 \\ 1.80 & 3.17 & 0.80 & 0 & 0 & 0 \\ 1.80 & 1.80 & 3.17 & 0 & 0 & 0 \\ 0 & 0 & 0 & 1.13 & 0 & 0 \\ 0 & 0 & 0 & 0 & 1.13 & 0 \\ 0 & 0 & 0 & 0 & 0 & 1.13 \end{bmatrix}$ |  |
| 40% | SIMP |  | $\begin{bmatrix} 1.08 & 0.64 & 0.64 & 0 & 0 & 0 \\ 0.64 & 1.08 & 0.64 & 0 & 0 & 0 \\ 0.64 & 0.64 & 1.08 & 0 & 0 & 0 \\ 0 & 0 & 0 & 0.46 & 0 & 0 \\ 0 & 0 & 0 & 0 & 0.46 & 0 \\ 0 & 0 & 0 & 0 & 0 & 0.46 \end{bmatrix}$ |  |
| | Grid point density |  | $\begin{bmatrix} 3.95 & 2.30 & 2.30 & 0 & 0 & 0 \\ 2.30 & 3.95 & 2.30 & 0 & 0 & 0 \\ 2.30 & 2.30 & 3.95 & 0 & 0 & 0 \\ 0 & 0 & 0 & 1.22 & 0 & 0 \\ 0 & 0 & 0 & 0 & 1.22 & 0 \\ 0 & 0 & 0 & 0 & 0 & 1.22 \end{bmatrix}$ |  |
| 50% | SIMP |  | $\begin{bmatrix} 4.10 & 2.45 & 2.45 & 0 & 0 & 0 \\ 2.45 & 4.10 & 2.45 & 0 & 0 & 0 \\ 2.45 & 2.45 & 4.40 & 0 & 0 & 0 \\ 0 & 0 & 0 & 1.52 & 0 & 0 \\ 0 & 0 & 0 & 0 & 1.52 & 0 \\ 0 & 0 & 0 & 0 & 0 & 1.52 \end{bmatrix}$ |  |
| | Grid point density |  | $\begin{bmatrix} 7.00 & 3.58 & 3.58 & 0 & 0 & 0 \\ 3.58 & 7.00 & 3.58 & 0 & 0 & 0 \\ 3.58 & 3.58 & 7.00 & 0 & 0 & 0 \\ 0 & 0 & 0 & 2.48 & 0 & 0 \\ 0 & 0 & 0 & 0 & 2.48 & 0 \\ 0 & 0 & 0 & 0 & 0 & 2.48 \end{bmatrix}$ |  |

Through the above cases, compared to the negative Poisson's ratio three-dimensional microstructure obtained by variable density, the three-dimensional structure obtained by this algorithm has similar configurations. Due to the introduction of the Heaviside smoothing function in the algorithm, a smooth structural boundary is formed, thereby avoiding the smooth processing of the structural boundary before printing. Due to the fast convergence and high efficiency of the SIMP framework itself, the topology algorithm,

based on grid point density and developed by the SIMP framework, has achieved a flexible and efficient optimisation process.

As the volume fraction increases, the bulk modulus of the microstructure increases, leading to an increase in the stiffness properties of the microstructure. The evolutionary history of the bulk modulus and volume fraction relative to the iteration when the volume fraction is 30%, as shown in Figure 9.

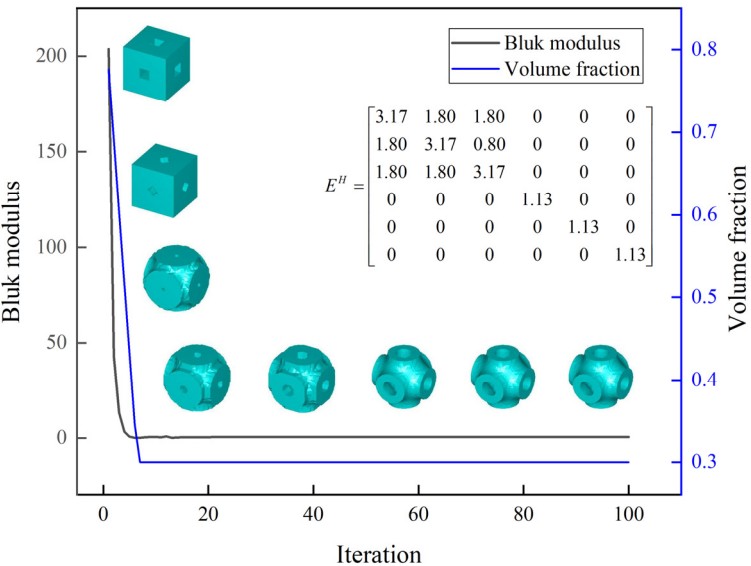

**Figure 9.** 30% volume fraction topology convergence history.

### 4.2. Compressive Mechanical Properties of 3D Internally Concave Surface Structures

In this paper, the 3D concave surface structure shown in Figure 10e is selected, and three sets of $5 \times 5 \times 5$ combined structures are designed by varying the wall thickness, with individual microstructure dimensions of 5 mm $\times$ 5 mm $\times$ 5 mm. 316L powder is selected for 3D printing. Using bidirectional powder spreading metal 3D printing technology, the maximum forming printing size of the printer is 800 mm $\times$ 700 mm $\times$ 1000 mm. Using a single-layer powder thickness of 0.03 mm and a laser wavelength of 1064 nm, the formed sample is placed in a universal testing machine for compression testing. Uniaxial compression experiments and finite element calculations are carried out on the internal concave surface structure, as shown in Figure 11; the finite element results are displayed as vertical displacement cloud maps. As this paper is only a verification analysis of the 3D bulk modulus maximisation topology, only the mechanical properties of the inner concave surface structure with different wall thicknesses are investigated. This paper encourages the secondary design of the structure based on the topology and its innovative application to engineering practice.

The consistency of the finite element model and experiment was verified by comparing the stress-strain curves, as shown in Figure 12. The structures with different wall thicknesses have the same stress-strain relationship in the initial stages of compression, with higher stress plateaus occurring as the wall thickness increases, and the slope of the stress-strain curve increases extremely at strains above 0.68, indicating a fully dense structure. Stress yielding occurs at the end of the linear elastic phase of the structure, followed by a degree of softening and stress fluctuations during the stress plateau phase. The stress plateau phase occurs as a result of the lattice structure yielding to damage, then deforming plastically at an almost constant pressure and generating energy absorption. The appearance of post-yielding (stress fluctuations) implies that structural instability has occurred in the structure at certain locations. As the wall thickness increases, the softening of the plateau at the 10–20% stage weakens and turns into hardening; this is due to the fact that the thicker Arc-shaped walls do not soften as the macroscopic deformation progresses, as the

inner walls meet earlier at the bending maximum, and a certain amount of hardening behaviour occurs. Post-yielding also weakens significantly for the same reason, while densification becomes more evident. In the compression strain range of 40–70%, significant stress fluctuations occur due to the local buckling of the curved shell wall. It can be seen, from the stress-strain curves in Figure 12, that none of the three models have yet entered a significant densification phase until they reach 70% strain. In summary, the stress plateau is relatively stable and homogeneous compared to common porous structures [61,62].

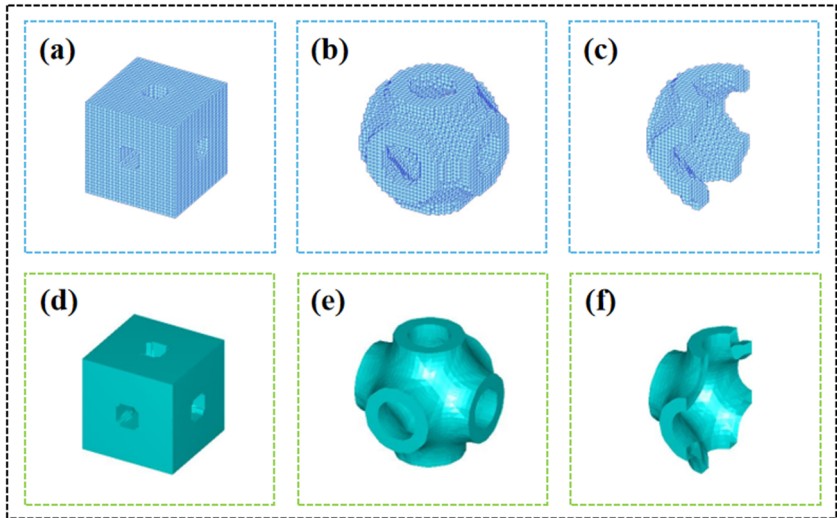

**Figure 10.** (**a**) Initial optimised design. (**b**) 3D internal concave surface structure. (**c**) Central section. (**d**–**f**) Initial optimised design after boundary homogenisation, 3D internal concave surface structure and central section.

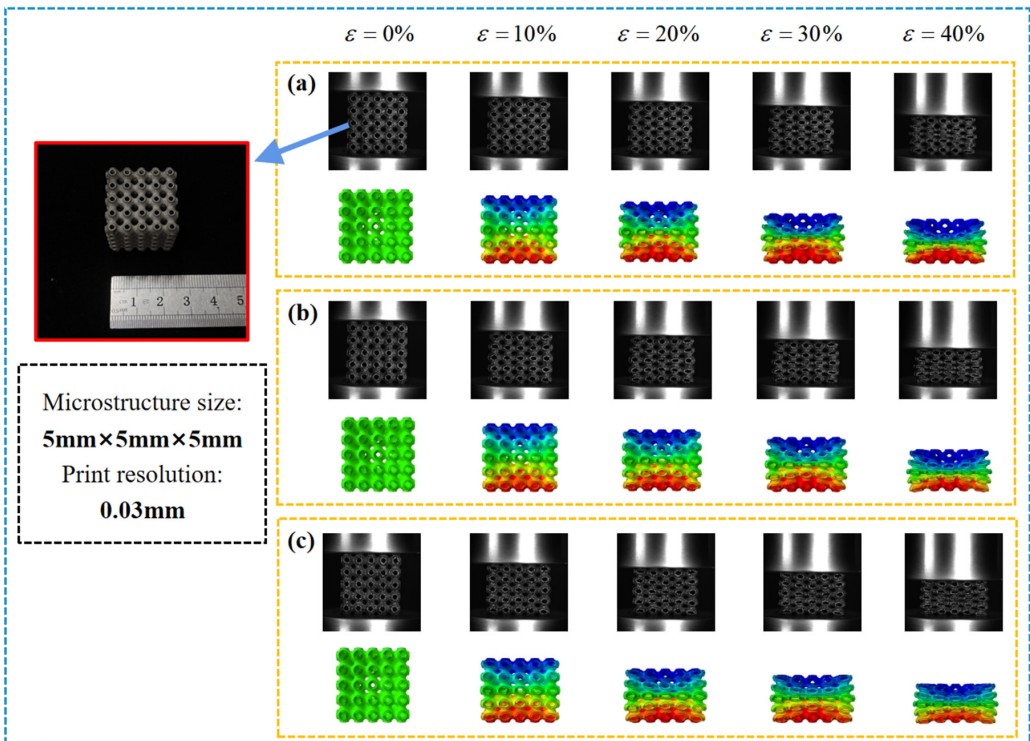

**Figure 11.** (**a**) Compression tests and finite element calculations for a concave curved structure with a wall thickness of 0.4 mm. (**b**) Wall thickness 0.5 mm internal concave surface structure. (**c**) Wall thickness 0.6 mm internal concave surface structure.

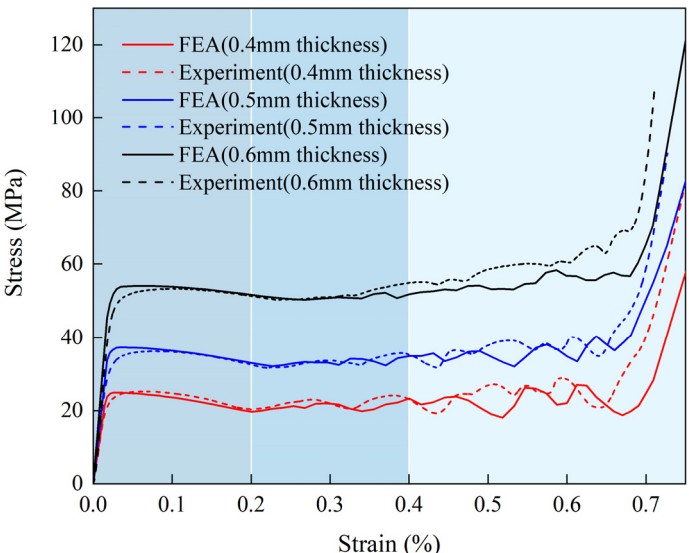

**Figure 12.** Stress-strain relationships for internally concave curved structures with different wall thicknesses.

In this paper, based on 3D printing technology, the resulting sample and the topology optimisation model are in good agreement, but the printing process results in a rough surface, as shown in Figure 13. This is due to the bonding of incompletely molten powder on the surface and the step effect, and is also the reason why the FEA results are higher than the experimental results. This type of inner concave surface structure has light mass, high specific strength, and high energy absorption capacity, and the 3D topology has a smooth surface and uniform radius of curvature after interpolation of the shape function, so the structure has a uniform stress distribution during load bearing, which has stronger fatigue service performance and broader application prospects.

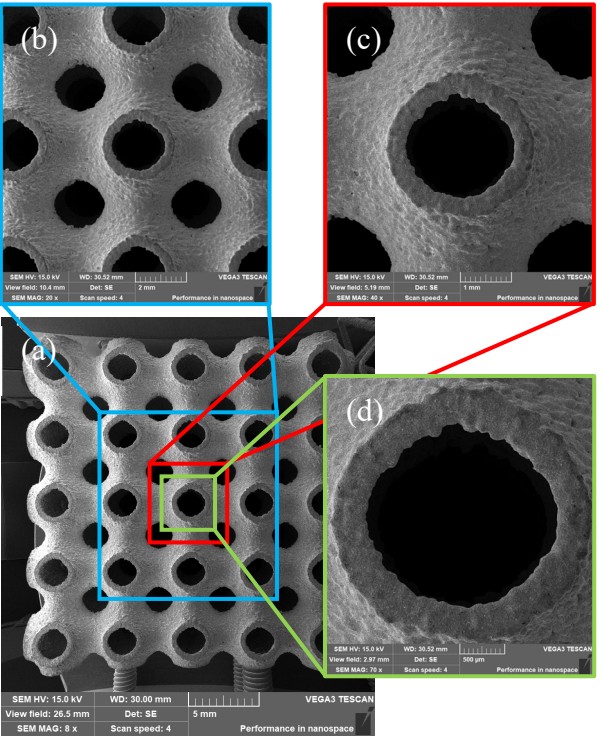

**Figure 13.** SEM image of the inner concave curved structure prepared in 316L powder. (**a**) View field 26.5 mm. (**b**) View field 10.4 mm. (**c**) View field 5.19 mm. (**d**) View field 2.97 mm.

## 5. Conclusions

This article is based on the homogenisation theory and establishes a topology optimisation model with grid point density as the penalty objective, aiming to maximise the bulk modulus. This method is used for topology optimisation design of periodic microstructure materials. The calculation results show that this method can achieve the maximum bulk modulus design of microstructure metamaterials and obtain a microstructure topology with smooth boundaries. Taking the optimisation objective of maximum bulk modulus as an example, the microstructure configuration can meet the optimisation objective and obtain the optimal solution in fewer iteration steps, which is consistent with the existing microstructure topology model. Second, this article proposes a method for calculating the parameters of plane internal forces, which can be extended to other microstructure configurations. It further demonstrates the effectiveness of topology optimisation by printing models with smooth boundaries and adding techniques, providing an effective method to achieve the design and special properties of artificial metamaterials. Finally, compression experiments were conducted on the concave surface microstructure of the topology based on 3D metal printing technology. The finite element simulation and experimental results showed good consistency. Through scanning electron microscopy, the partially melted powder adhered to the surface of the printed structure was displayed, and the entire process analysis of the topology microstructure from design to forming under additive manufacturing was further analysed.

The topology optimisation of microstructures achieves the design of microstructures with special properties. A single microstructure can meet the target requirements. However, the application of microstructures requires a macroscopic environment after periodic combination of cells, and the combined macroscopic structure often loses the mechanical properties (maximum stiffness, negative Poisson's ratio) possessed by a single cell. In the future, topology optimisation of microstructures requires further optimisation of structural design to meet macroscopic requirements, Alternatively, additional target requirements for special attributes (vibration isolation, thermal insulation) can be added to achieve a microstructure topology with multiple physical fields.

**Author Contributions:** Conceptualisation, X.Z. and X.L.; methodology, X.Z.; software, X.Z. and C.T.; validation, X.Z. and C.T.; formal analysis, X.Z. and C.T.; investigation, X.Z.; resources, X.L.; data curation, X.Z.; writing—original draft preparation, X.Z.; writing—review and editing, X.L. and C.T.; visualization, X.Z.; supervision, H.L.; project administration, Z.L.; funding acquisition, X.L. All authors have read and agreed to the published version of the manuscript.

**Funding:** This work was funded by the National Natural Science Foundation of China grant number (11902287) and the Research Program of Ministry of Science and Technology of China grant number (1816300104 and 2016300402).

**Data Availability Statement:** The data used to support the findings of this study are available from the corresponding author (ysulxi@163.com) upon request.

**Conflicts of Interest:** The authors declare that there are no conflicts of interest regarding the publication of this paper.

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
