# Peer review of "Design and Mechanical Properties of Maximum Bulk Modulus Microstructures Based on a Smooth Topology with Grid Point Density"

_aerospace, doi:10.3390/aerospace11020145_

Round 1
Reviewer 1 Report
Comments and Suggestions for Authors
The manuscript describes approach of topology optimization which could be useful for additive manufacturing. It is still urgent task and the provided study helps to expand practical value and scope of AM techniques. However, there are some comments concerning experimental part of the research, which has received too little attention. Please, revise the manuscript taking into account following comments, questions and suggestions:
1. Some more details about experimental part of the work should be added in abstract and introduction, in my opinion.
2. Also, more detailed information about experiment should be provided in the Section 4.2, including description of the used equipment, printing parameters, etc.
3. What do the colors of calculated models in Figure 9 mean?
4. How did you find or calculate print resolution value? Did you make cross-sections of the printed samples and measure the actual wall thicknesses?
Reviewer 2 Report
Comments and Suggestions for Authors
Your work is indeed noteworthy and contributes significantly to the field. I appreciate the clarity of your methodology and findings. The research has great potential, and I believe a few enhancements could further elevate the manuscript. Here are some suggestions that might enhance the overall quality and impact of your work:
- Please provide a more comprehensive analysis in the Discussion section.
- Compare the performance of the proposed topology optimisation model against previously reported model(s).
- Highlight the strengths and weaknesses of the proposed model in comparison to others
- Discuss how the results align with existing literature, theories, or expectations.
- Address limitations of the study openly and suggest avenues for future research that could address these limitations.
- Recommend including a section on future work, addressing potential areas for further exploration and research.
Round 2
Reviewer 1 Report
Comments and Suggestions for Authors
I'm satisfied with given comments and corrections made. Thank you for thorough revision.
I would recommend accepting the manuscript.